# Unveiling the Mechanism of Action of Palmitic Acid, a Human Topoisomerase 1B Inhibitor from the Antarctic Sponge *Artemisina plumosa*

**DOI:** 10.3390/ijms26052018

**Published:** 2025-02-26

**Authors:** Alessio Ottaviani, Davide Pietrafesa, Bini Chhetri Soren, Jagadish Babu Dasari, Stine S. H. Olsen, Beatrice Messina, Francesco Demofonti, Giulia Chicarella, Keli Agama, Yves Pommier, Blasco Morozzo della Rocca, Federico Iacovelli, Alice Romeo, Mattia Falconi, Bill J. Baker, Paola Fiorani

**Affiliations:** 1Department of Onco-Hematology, Gene and Cell Therapy, Bambino Gesù Children’s Hospital-IRCCS, Via Ferdinando Baldelli 38, 00146 Rome, Italy; alessio.ottaviani@opbg.net; 2Department of Biology, University of Rome Tor Vergata, Via della Ricerca Scientifica, 00133 Rome, Italy; davide.pietrafesa@uniroma2.it (D.P.); drbinicsoren@gmail.com (B.C.S.); jagadishdrdasari@gmail.com (J.B.D.); beatrice.messina702@gmail.com (B.M.); francesco.demofonti@hotmail.com (F.D.); giuliachica57@gmail.com (G.C.); blasco.morozzo.della.rocca@uniroma2.it (B.M.d.R.); federico.iacovelli@uniroma2.it (F.I.); alice.romeo@uniroma2.it (A.R.); falconi@uniroma2.it (M.F.); 3Department of Chemistry, University of South Florida, USF Sweetgum Ln 12111, Tampa, FL 33620, USA; olsens@usf.edu (S.S.H.O.);; 4Laboratory of Molecular Pharmacology, Center for Cancer Research, National Cancer Institute, Convent Drive 37, Bethesda, MD 20892, USA; agamak@mail.nih.gov (K.A.); yves.pommier@nih.gov (Y.P.); 5Institute of Translational Pharmacology, National Research Council, CNR, Via del Fosso del Cavaliere 100, 00133 Rome, Italy

**Keywords:** human DNA topoisomerase, Antarctic extracts, palmitic acid, molecular dynamic

## Abstract

Cancer remains a leading cause of death worldwide, highlighting the urgent need for novel and more effective treatments. Natural products, with their structural diversity, represent a valuable source for the discovery of anticancer compounds. In this study, we screened 750 Antarctic extracts to identify potential inhibitors of human topoisomerase 1 (hTOP1), a key enzyme in DNA replication and repair, and a target of cancer therapies. Bioassay-guided fractionation led to the identification of palmitic acid (PA) as the active compound from the Antarctic sponge *Artemisina plumosa*, selectively inhibiting hTOP1. Our results demonstrate that PA irreversibly blocks hTOP1-mediated DNA relaxation and specifically inhibits the DNA religation step of the enzyme’s catalytic cycle. Unlike other fatty acids, PA exhibited unique specificity, which we confirmed through comparisons with linoleic acid. Molecular dynamics simulations and binding assays further suggest that PA interacts with hTOP1-DNA complexes, enhancing the inhibitory effect in the presence of camptothecin (CPT). These findings identify PA as a hTOP1 inhibitor with potential therapeutic implications, offering a distinct mechanism of action that could complement existing cancer therapies.

## 1. Introduction

DNA topoisomerases are crucial enzymes in maintaining the topological states of DNA during various cellular processes, such as replication, transcription, and chromosomal segregation [1]. These enzymes create transient single-strand breaks in the DNA molecule, allowing the relaxation of supercoiled DNA to relax and facilitating the necessary unwinding for essential genetic activities [2].

Among the two main classes of DNA topoisomerases, type I and type II, DNA topoisomerase I (TOP1) is unique in its ability to relieve torsional stress without requiring ATP, making it an energy-efficient catalyst in managing DNA topology. The specific isoform TOP1 exhibits a broad range of functionality across different organisms, from prokaryotes to eukaryotes, highlighting its evolutionary importance [3]. TOP1’s mechanism of action involves the formation of a covalent intermediate with the DNA, enabling the rotation of the DNA strands around the break site [4]. This precise control of DNA dynamics is essential for preventing harmful DNA supercoiling, which can impede the progress of replication forks and transcriptional machinery.

Researchers have underscored the critical role of human TOP1 (hTOP1) in genomic stability and its potential as a therapeutic target [5]. This enzyme is a 91kDa protein of 765 amino acids divided in four domains: the N-terminal domain (1–214), the core domain (215–635), the linker (636–712), and the C-terminal domain (713–765) containing Tyr 723, which undergoes the nucleophilic attack to the substrate and forms together with Arg 488, Lys 532, Arg 590, and His 632 at the catalytic site [6].

HTOP1’s involvement in regulating DNA supercoiling and resolving topological issues are vital for proper cell function and division [7]. Dysregulation of hTOP1 activity has been linked to various pathological conditions, including cancer, where overexpression or mutation of the enzyme can lead to genomic instability and tumorigenesis [8]. Consequently, enzyme inhibitors have emerged as promising candidates in anticancer therapy, aiming to exploit the enzyme’s pivotal role in DNA metabolism to induce cytotoxicity specifically only in cancer cells [9]. Among them we can enumerate camptothecin (CPT) an anticancer drug that targets only hTOP1 and is derived from the bark of the Chinese plant *Camptotheca acuminata* [10]. CPT exerts its anticancer effects by stabilizing the hTOP1-DNA cleavable complex, thereby preventing DNA religation and causing the accumulation of DNA damage [11]. This leads to replication fork collapse, activation of DNA damage responses, and, ultimately, cell death [12,13]. Due to their rapid division, cancer cells are more susceptible to being killed by hTOP1 inhibitors than normal cells [14].

Starting from CPT, several derivatives were developed that are now currently used in clinical settings, such as irinotecan and topotecan, for treating various cancers [15]. Akin to what happened for CPT, whose discovery as a hTOP1 poison was a serendipitous accident, nowadays there are still numerous natural compounds that remain to be discovered and could lead to the development of new and more effective anticancer drugs [16]. In this context, cold-water organisms in Antarctica are relatively unexplored in terms of their biomedical potential [17]. This is mainly due to the difficulty in accessing these environments. Still, it was also historically driven by the long-held perception that low biodiversity would result in low chemo diversity [18,19]. A description of polar regions, which encompass 30% of global geography, as the source of less than 3% of published marine natural products, renewed interest in polar bioprospecting and has led to impressive advances [20].

The current paradigm recognizes that marine organisms in such harsh environments require unique behavioral, physical, and chemical defense mechanisms to survive [21], the latter of which results in natural products with promising anticancer, anti-inflammatory, and antibacterial activity [19].

Our interest in discovering bioactive metabolites from Antarctic invertebrates led us to screen an extensive library comprising more than 750 chemical extracts for inhibiting the catalytic activity of hTOP1. The invertebrates primarily included sponges, corals, tunicates and mollusks, and were all hand-collected by SCUBA divers in the vicinity of Palmer Station, Antarctica (64°46.345′ S, 64°02.915′ W). In particular, an extract from the sponge *Artemisina plumosa* was active on this screen. Through successive rounds of purification, driven by a bioassay-guided approach, the active component of the sponge was identified as being palmitic acid (PA). To respect ecological concerns and minimize environmental impact on the Antarctic region, we did not collect additional samples of *Artemisina plumosa* after identifying PA as the active component. Instead, we opted to use commercially available PA for further studies.

Although it was already reported that PA can act as a hTOP1 inhibitor with antitumor characteristic [22,23], its mechanism of action still needs to be elucidated. Here we describe an in vitro and in silico approach, describing PA mechanism of action of inhibiting hTOP1 and its potential as lead molecule for developing an anticancer drug.

## 2. Results

### 2.1. Antarctic Invertebrate Extracts Inhibit hTOP1 Relaxation of Supercoiled DNA

The 58 crude extracts tested, which were representative of the 750 extracts initially screened, were derived from different Antarctic invertebrates and evaluated for their inhibitory effect on hTOP1 activity. Specifically, 10 µL of each crude extract was incubated with supercoiled plasmid DNA and hTOP1 for 1 h. The reaction was then stopped using 1X SDS stop dye, as described in the Methods section. The samples were subsequently analyzed on an agarose gel and stained with EtBr. As shown in Appendix A, among all the screened samples, only organism 10 (Appendix A, lane 10) resulted in complete inhibition of hTOP1 relaxation activity. In contrast, organisms 25, 37, 42, and 58 demonstrated slight inhibition of the enzyme (Appendix A, lanes 25, 37, 42, and 58). As a control, supercoiled DNA was incubated with the same amount of DMSO but without the enzyme (Appendix A, lane 59). The organism represented in Appendix A, lane 10, corresponds to a crude extract derived from *A. plumosa*. Starting with this crude extract, we performed several cycles of HPLC purification to isolate the compound responsible for hTOP1 inhibition, which was identified as PA.

### 2.2. PA Inhibits the Catalytic Activity of hTOP1

The inhibitory effect of PA (Figure 1A) on hTOP1 activity was assessed by a dose-dependent relaxation assay (Figure 1B). Purified protein was incubated with a DNA supercoiled plasmid in the absence or presence of an increasing concentration of PA for 1 h. Samples were analyzed by electrophoresis on agarose gel. The obtained results indicated that PA inhibited the relaxation activity of hTOP1 in a dose-dependent manner. Indeed, adding PA to the hTOP1 and DNA determined the inhibition of the relaxation activity. The presence of supercoiled DNA is detectable at a concentration of 100 µM (Figure 1B, lane 6). It rises with increasing dosage reaching clear evidence of supercoiled DNA at 200 µM (Figure 1B, lane 8). PA did not affect the electrophoretic mobility of DNA in the absence of hTOP1 at 200 µM (Figure 1B, lane 9). Since PA was dissolved in DMSO, we assayed the activity in the presence of an identical amount of DMSO without PA, demonstrating that DMSO does not affect the relaxation of hTOP1 (Figure 1B, lane 1). The band corresponding to supercoiled DNA was quantified and plotted as a function of PA concentration, as shown in Appendix A.

Relaxation assays were also performed as a time course experiment to evaluate whether PA inhibits hTOP1 reversibly or irreversibly. We used 100 µM of PA, as it was the lowest concentration showing inhibition in the supercoiled relaxation assay. Additionally, we included CPT at 100 µM, a well-known reversible inhibitor of hTOP1. Supercoiled DNA was incubated with either 100 µM PA, 100 µM CPT, or the same amount of DMSO as a control. After adding hTOP1, incubation was started at selected time points, as shown in Figure 1C, and samples were collected at different intervals ranging from 0.5 min to 60 min, representing various stages of enzyme inhibition. The samples were then analyzed by agarose gel electrophoresis. As shown in Figure 1C, in the presence of DMSO, hTOP1 exhibited complete relaxation of supercoiled DNA (Figure 1C, lanes 1–6). When incubated with PA, a lasting inhibitory effect was observed over time (Figure 1C, lanes 7–12), suggesting that PA’s inhibition of hTOP1 catalytic activity is irreversible within the time range explored (up to 60 min). On the other hand, CPT (Figure 1C, lanes 13–18) exhibited the typical behavior of a reversible inhibitor. Despite an initial inhibition of hTOP1 activity (Figure 1C, lanes 13 and 14), enzyme function gradually recovered over time, consistent with CPT’s well-documented reversible mechanism of action.

As an additional support of the irreversibility of PA inhibition, we performed a dilution experiment. The complete inhibition of the PA–enzyme also remains after a 3-fold or 9-fold dilution, indicating that the inhibitory effect is irreversible as shown in Appendix A lanes 4–6 where it is also shown, as a control, that the enzyme incubated with DMSO, and diluted maintains its activity (Appendix A lanes 1–3).

To demonstrate that the inhibition of hTOP1 by PA is not a general property of all fatty acids, we incubated purified protein with a supercoiled plasmid in the absence or presence of increasing concentrations of linoleic acid (LA) (Appendix A) for 60 min. LA is a fatty acid, with the chemical formula C_18_H_32_O_2_, contains two double bonds in its carbon chain while PA is a saturated fatty acid with the chemical formula C_16_H_32_O_2_ (Appendix A). The reported dose-dependent relaxation assay with increasing concentration of LA (Appendix A), shows that the absence of MSC DNA and the presence of several topoisomers, depicting unwinding of supercoiled DNA, confirm that this fatty acid does not inhibit hTOP1.

### 2.3. PA Inhibits hTOP1-Mediated DNA Religation

Once we assessed the minimal dose inhibiting hTOP1 and that the PA behavior is irreversible, we performed religation kinetic assays to investigate which step of the catalytic cycle of the enzyme is affected by PA using the substrate indicated in Figure 2A. Briefly, we incubated 0.6 pmol of the enzyme with 1.2 pmol of suicide substrate to form the cleavage complex (Figure 2B, lane 2). The reaction mixture was then split into two tubes, with 100 µM PA in one tube and the same amount of DMSO into the other. A 200-fold molar excess of complementary R11 oligonucleotide (5′-AGAAAAATTTT-3′) was added to initiate the religation process. Aliquots were taken at various time points, ranging from 1 to 15 min, and the reactions were stopped with 0.5% SDS. Products were analyzed by denaturing PAGE (Figure 2B). The results show that PA inhibits the religation step of the enzyme’s catalytic cycle (Figure 2B, lanes 7–10), while DMSO alone allowed proper religation of the R11 oligo (Figure 2B, lanes 3–6). A plot of the percentage of religated product is shown in Figure 2C.

### 2.4. HTOP1–DNA Cleavage Complex Reversal Assay

The hTOP1–DNA cleavage complex reversal assay is a biochemical assay used to study the activity of hTOP1 and its interaction with DNA. This assay is essential for elucidating the mechanisms of TOP1 inhibitors. The stability of the covalent DNA–enzyme complex was analyzed using a double-stranded DNA substrate, labeled with 6-fluorescein amidite (FAM) at one of the 3′-ends (Figure 3A). DNA was incubated with hTOP1 and 1µM of CPT (Figure 3B lanes 2–8) or 10 µM PA (Figure 3B lanes 9–15) or both CPT and PA (Figure 3B lanes 16–22), and DMSO as control (Figure 3B lanes 23–29). An amount of 0.35 M NaCl was added to induce the reversal of the hTOP1 cleavage complexes. Aliquots were taken, and reactions terminated at various time points: 0, immediately before NaCl addition (Figure 3B lanes 2, 9, 16 and 23), 0.5 min, 1 min, 2 min, 5 min, 10 min, and 20 min. The representative gel in Figure 3B shows that PA alone did not inhibit the religation phase of hTOP1 (lanes 9–15). This inhibition occurred only when CPT and PA were incubated together (lanes 16–22). We also tested higher concentration of PA, but none affected the stabilization of the cleavable complex (Appendix A).

### 2.5. Pre-Incubation Dose-Dependent Assay

We also performed a relaxation assay pre-incubating the hTOP1 with PA, before adding the substrate (Figure 4). In this experiment the enzyme was incubated with 150 μM and 200 μM PA for 5 min at 37 °C before adding the supercoiled DNA. The reaction mixture was further incubated for 30 min. At the same time, we incubated supercoiled DNA with PA simultaneously, then we added hTOP1 to start the reaction. As control, we added the same amount of DMSO as PA to the reaction mixture to evaluate any possible interaction of the PA solvent to the substrate (Figure 4 lane 1). At the end of the experiment 0.5% of SDS was added to all samples that were loaded on an agarose gel. Figure 4 reported the effect of pre-incubating PA with the enzyme. In the pre-incubation setting, the enzyme was able to relax the supercoiled DNA completely (Figure 4 lanes 2–3), while in the simultaneous condition, the enzyme is affected by PA due to the presence of supercoiled DNA at the bottom of the gel (Figure 4 lanes 4–5) as negative control (Figure 4 lane 6).

### 2.6. Computational Analyses

To elucidate the molecular mechanism of action of PA on hTOP1, we computationally modelled three distinct systems, which were subjected to extensive classical molecular dynamics (MD) simulations: (1) hTOP1 in complex with a double-stranded DNA (dsDNA) substrate, serving as a reference system; (2) a ternary complex comprising hTOP1, dsDNA, and the known inhibitor CPT; and (3) a system mimicking our in vitro experimental conditions, wherein PA molecules were introduced to the hTOP1-DNA complex at a defined stoichiometry. This comparative approach facilitated the investigation of PA’s influence on hTOP1-DNA interactions in relation to the established effects of CPT.

### 2.7. Principal Component Analysis

To investigate the impact of ligand binding on the conformational dynamics of hTOP1, we analyzed the collective motions of the simulated systems. Specifically, we focused on the essential motions captured by the first eigenvector (PC1), which represents the largest-amplitude motions within the protein structure. These results are visualized in Figure 5A, where it can be observed that ligand binding significantly enhanced hTOP1 flexibility, particularly within the linker (residues 636–712) and core (residues 59–479) domains. This increased mobility was observed in both the hTOP1-DNA-CPT and hTOP1-DNA-CPT-PAs complexes compared to the hTOP1-DNA complex.

Analysis of the 2D projections of the first two principal components of motions (Figure 5B) shows reduced conformational sampling in the hTOP1-DNA complex (left) compared to the hTOP1-DNA-CPT (center) and hTOP1-DNA-CPT-PAs (right) systems. The latter exhibits a broader distribution, particularly evident in the hTOP1-DNA-CPT-PAs system. This can be linked to the sampling of conformations different from those compatible with a functional protein.

### 2.8. MM/PBSA

MM/PBSA analyses were used to estimate the interaction energy between the CPT and the hTOP1-DNA complex in the presence and absence of the PAs ligands, as shown in Table 1. These analyses revealed a strong interaction between CPT and hTOP1-DNA in both systems, confirming the ability of CPT to intercalate in the site of the DNA cleavage, thus inhibiting the religation step. Interestingly, the analyses highlighted a lower energy value in the hTOP1-DNA-CPT-PAs complex (−43.45 ± 3.39 kcal/mol). Although both Van der Waals (VdW) and electrostatic interactions contribute substantially to the overall binding energy, VdW forces play a more dominant role. Moreover, the hTOP1-DNA-CPT-PAs complex displayed the highest contributions from both VdW and electrostatic interactions, suggesting that PA molecules enhance the binding of CPT.

### 2.9. Computational Analyses of DNA-PA Systems

A system composed of linear DNA and ten PA molecules was built and simulated through Gaussian accelerated Molecular Dynamics (GaMD) to computationally mimic the experimental conditions of the hTOP1 religation assay. Visual inspection of three MD replicates of the DNA-PA systems suggests the mechanism by which the PA molecules interact with the DNA over the course of a 250 ns simulation. The PA molecules progressively engage with the DNA strand during the simulations, forming stable contacts. This interaction leads to a structured complex where the hydrophobic tails of the PA molecules orient themselves towards the DNA bases, creating a likely hydrophobic core. Concurrently, the polar environment plays a key role in orienting the charged moieties, with the negatively charged heads of the PA molecules and the phosphate groups of the DNA backbone facing the surrounding aqueous environment. This type of arrangement can be observed across all replicates. Representative snapshots of these structural rearrangements of the first replica are shown in Figure 6. Furthermore, the analysis of the radius of gyration (Rg), a quantitative measure of the system’s compactness, confirms these observations.

The Rg was calculated for the entire DNA-PA complex over the course of the 250 ns simulation (Appendix A). A significant reduction in the Rg occurs from 75 ns onwards, which reflects an increase in the compactness of the complex. In addition, the complex’s buried surface area (BSA) was computed (Appendix A). The BSA represents the surface area of the complex that becomes inaccessible to the solvent upon formation of the DNA-PA structure. A consistent increase in BSA during the simulation shows that larger portions of the PA and DNA molecules become buried within the complex, further confirming the formation of a stable, compact structure.

## 3. Discussion

Cancer continues to be one of the top causes of death globally, driving the ongoing search for more effective treatments. Therefore, there is a need to discover new antitumor compounds that not only exhibit potent therapeutic effects but also minimize adverse reactions. With their structural diversity and biological activity, natural products present a promising reservoir for such novel compounds. Exploring the vast chemical diversity found in plants, marine organisms, and microorganisms could lead to identifying innovative and more tolerable anticancer drugs. In this study, we performed a comprehensive screening of 750 Antarctic extracts to identify potential inhibitors of hTOP1. Among all the tested extracts, a bioassay-guided fractionation procedure was used to isolate and identify PA as the active compound inhibiting hTOP1 from the Antarctic sponge *Artemisina plumosa*. This struck our interest since PA has been associated with topoisomerase inhibition. Our results are consistent with previous studies. Karna et al. demonstrated that PA inhibits hTOP1 relaxation of a supercoiled DNA [22]. They tested hTOP1 inhibition on human non-small cell lung cancer cell line A549 and evaluated the cytotoxicity of PA, with MTT-based viability assay, determining the IC_50_ of PA to be around 150 μM [22]. These results were confirmed by Harada et al., who described not only the antitumor activity of PA on human T cell acute lymphoblastic leukemia cell line MOLT-4, but also its selectivity against hTOP1 without affecting hTOP2 enzymes [23]. However, the molecular mechanism through which PA inhibits hTOP1 activity had not been elucidated. Here, we report experiments aimed at understanding the mechanism of action responsible for inhibiting this enzyme.

The resulting picture suggests a dynamic behavior of PA, where context-dependent interactions appear to drive the functional outcome. In detail, our results showed that PA effectively inhibits the activity of hTOP1 in relaxing supercoiled DNA (Figure 1B). Moreover, the results of the time course relaxation assays suggest that enzyme inhibition by PA is irreversible (Figure 1C), at least in the time frames assayed. Additionally, the inhibition also remains after a 9-fold dilution, at a concentration where the enzyme, in the absence of the drug, is active (Appendix A), confirming that the PA inhibition is irreversible.

To further evaluate that this inhibition was not a general feature of fatty acids, we also tested LA in plasmid relaxation assays (Appendix A). This 18-carbon polyunsaturated fatty acid did not inhibit hTOP1, highlighting the specificity of PA’s inhibitory action, and suggesting that the structure and saturation level of fatty acids play a role in their interaction with hTOP1. To study the mechanism of PA inhibition, we employed religation assays to determine whether PA affects the DNA rejoining step of the hTOP1 catalytic cycle. Through this assay we evaluated that, in presence of PA, the enzyme could not complete its catalytic cycle and that the religation of the complementary strand was strongly inhibited. In addition, to assess the stability of the cleavable complex in the presence of PA, we also performed cleavage complex reversal assays, where we found that PA enhances the trapping of TOP1 by CPT, but does not lead to the formation of stable cleavable complexes by itself (Figure 3), even at high doses (Appendix A). The cleavage complex reversal experiments shown in Figure 3 (lanes 9–15), regarding the action with PA alone, appear to be in contrast with the religation assays (Figure 2). Although the results from Figure 2 show that PA inhibits religation when using the suicide substrate, this inhibition is not observed in the cleavage reversal assay in the presence of PA alone (Figure 3, lane 9–15). This discrepancy could be explained by differences in both the conditions and mechanisms underlying these two assays. The suicide substrate directly mimics the intermediate step of the religation process, where PA may specifically interfere with the enzyme’s ability to religate the DNA strands. In contrast, the cleavage complex reversal assay involves both the formation and then the resolution of a stable cleavage complex. PA may not stabilize this complex sufficiently on its own to block the reversal process, as it does when it is combined with CPT (Figure 3, lanes 16–22). Therefore, while PA efficiently blocks religation in the suicide substrate assay, it might not generate a persistent cleavage complex on its own. Based on this conjecture, we suggest that PA needs a protein that is covalently attached to the DNA in order to exert its inhibitory effect. Indeed, when PA is added together with the DNA and the protein, we do not observe any stabilization of the cleavage complex (Figure 3, lanes 9–15).

While we have discussed the hypothesis that PA requires the enzyme to be covalently attached to DNA to exert its inhibitory effect, we cannot exclude the possibility that, given the specifics of PA, it may also interfere with the cleaved single-stranded DNA fragments, making them less available for the religation process. This “masking effect” could reduce the effective concentration of free DNA ends necessary for religation, thus contributing in an additional way to the inhibition. To better clarify this point, we performed PA pre-incubation relaxation assays. In this pre-incubation experiment (Figure 4), PA is pre-incubated with the enzyme without any possible interaction with the DNA substrate. Intriguingly, PA did not have a measurable inhibiting effect (Figure 4, lanes 2,3), while an incubation of PA with the DNA resulted in the hTOP1 inhibition, suggesting that PA could partially shield the DNA, rendering it less accessible to the enzyme, preventing binding of the DNA, and thereby inhibiting its activity (Figure 1).

Although the explanations proposed above reconcile many aspects of our results, it must be noted, however, that the inhibition we observe in the relaxation of plasmid DNA when incubating PA with DNA is difficult to fit with all the results. A possible explanation could lie in the different conditions and substrates used in the assays: in one case, we are working with a supercoiled plasmid, while in the other with a double-stranded oligonucleotide. Moreover, the relaxation of supercoiled DNA is a more complex process that involves the entire enzymatic cycle, including association, cleavage, religation, and dissociation.

In silico analyses provide valuable insights into this complex behavior. Predictive modeling of PA binding sites on hTOP1 revealed three potential interaction sites, located both in proximity to and distant from the catalytic cleft (Appendix A). Some aspects of the inhibitory action of PA were further elucidated through molecular dynamics investigation, namely complexes of hTOP1, DNA and PA, and/or CPT. When PA or CPT molecules are added to the system, the MD simulations highlight a significant alteration in the collective motions of hTOP1 compared to the control system (hTOP1-DNA complex). This is evident in both the hTOP1-DNA-CPT and hTOP1-DNA-CPT-PAs complexes. The potential additive inhibitory effect, promoted by the PA molecules in the presence of CPT, can be observed in the broader distribution of the two-dimensional projections compared with the native motions of the complex. Furthermore, the binding free energy of CPT to the complex is enhanced in the hTOP1-DNA-CPT-PAs system when compared to the ternary complex (Table 1), lending support for the synergistic effect observed in the cleavage complex reversal assay.

GaMD simulations revealed that PA interacts with DNA to form a distinct complex characterized by the hydrophobic tails of PA contacting the DNA bases, creating a hydrophobic core, while the charged head groups and the phosphates are all oriented towards the solvent (Figure 6).

These findings suggest an alternative mechanism by which PA may inhibit complex formation. The identification of PA as an irreversible hTOP1 inhibitor can have significant implications. Unlike CPT and its derivatives [21,22], well-established hTOP1 inhibitors used in cancer therapy but with several side effects, PA offers a mechanism of action that could be exploited for new therapeutic benefits, both directly and as an adjuvant component for mixed therapies. Future research will focus on detailed mechanistic studies to fully elucidate the complex interactions between PA and DNA, as well as to investigate the role of the N-terminal domain of hTOP1 from a biochemical perspective. This will complement the in silico analysis conducted with the N-terminal truncated protein and further enhance our understanding of PA’s mechanism of action. Our findings lay the groundwork for further investigation into the therapeutic potential of PA and its derivatives.

## 4. Materials and Methods

### 4.1. Reagents

For agarose-based assays and the religation experiment, recombinant hTOP1 protein (Cat. No. ENZ-306) was purchased from Prospec (Hamada St. 8, Rehovot, Israel). PA was purchased from Merck and SDS was purchased from Sigma-Aldrich (St. Louis, MI, USA).

For hTOP1–DNA cleavage complex reversal assay, recombinant hTOP1 was purified from baculovirus as previously described [24]. NMR spectra were acquired using a Bruker Neo 600 MHz broadband spectrometer (Appendix A). The residual solvent peaks were used as an internal chemical shift reference (CDCl_3_: δ_C_ 77.0; δ_H_ 7.27). High-resolution mass spectrometry-liquid chromatography data were obtained on an Agilent 6540 LC-MS QTOF (Agilent Technologies, Santa Clara, CA, USA) coupled to an Agilent Jet-stream electrospray ionization detector. H_2_O (A) and 0.1% FA in CH3CN (B) were used as mobile phases on a Phenomenex Kinetex C18 column (2.6 mm, 100 Å, 150 × 3 mm: 0.5 mL/min). Reverse phase high-performance liquid chromatography (RF-HPLC) was performed on a Shimadzu LC20-AT system equipped with a photodiode array detector (M20A) using a preparative Phenomenex C18 column (5 mm, 100 Å, 250 × 21.2 mm: 10 mL/min) or a semi-preparative Phenomenex C18 column (10 mm, 100 Å, 250 × 10 mm: 4 mL/min). The methanol and acetonitrile used for column chromatography were obtained from Fisher Scientific (Thermo Fisher Scientific, Waltham, MA, USA) and were HPLC grade (>99% purity) while the H_2_O was distilled and filtered.

### 4.2. Bioassay-Guided Fractionation of PA from Artemisina plumosa

A bioassay-guided fractionation was performed on the sponge *Artemisina plumosa*, which was collected in 2016 near Palmer Station, Antarctica. The freeze-dried sponge (1.97 kg) was extracted in MeOH twice overnight. The extract was loaded onto HP20 resin by passing the extract through the column before diluting the eluent with an equal volume of H_2_O and passing it back through the column twice. The column was eluted with (1) 30% Me_2_CO/H_2_O, (2) 75% Me_2_CO/H_2_O, and (3) Me_2_CO. The three fractions were submitted for initial screening. The third fraction (1.55 g) showed inhibition for hTOP1 and was further purified with RP-HPLC. The fractions were further screened and fractionated using RP-HPLC until the pure bioactive compound was isolated, resulting in PA (10.4 mg).

### 4.3. Dose-Dependent and Time Course Relaxation Assays

The minimal inhibiting dose of PA on hTOP1 was determined using a dose-dependent relaxation assay of negatively supercoiled DNA pBlueScript KSII (-). The reaction, conducted in a 30 μL volume, included a buffer with 20 mM Tris-HCl pH 7.5, 0.1 mM EDTA, 10 mM MgCl2, 50 μg/mL acetylated bovine serum albumin (TOPO mix 1X), 150 mM KCl, 1 U of purified hTOP1, 0.5 μg pBlueScript, and varying PA concentrations. A positive control involved the enzyme with an equivalent amount of DMSO. After 1 h at 37 °C, the reaction was stopped using 0.5% SDS stop dye. To evaluate the reversibility or irreversibility of PA, the same experiment was carried out with 100 μM of PA, 100 μM of CPT or an equivalent amount of DMSO as a function of time, and reactions were stopped at the indicated time points with a final concentration of SDS 0.5%. Samples were resolved on 1% agarose gel in TBE 1X buffer and visualized with ChemiDoc Biorad (Bio-Rad Laboratories, Hercules, CA, USA) after staining with 0.5 μg/mL EtBr and destaining in dH_2_O. The same procedure for the dose-dependent assay was followed for the pre-incubation experiment, except that before adding the DNA, hTOP1 was pre-treated for 5 min at 37 °C with two concentrations of PA 150 μM and 200 μM. The reactions were stopped at different time points by adding SDS. All samples were resolved on a 1% agarose gel using 1X TBE buffer (48 mM Tris, 45.5 mM boric acid, 1 mM EDTA). The enzyme’s capacity to relax supercoiled DNA was visualized under a UV transilluminator following staining with 0.5 μg/mL EtBr and destaining with dH_2_O.

The irreversibility of PA inhibition was monitored by incubating hTOP1 and the inhibitor at 37 °C for 10 min. Following this, the inhibitor–hTOP1 mixture was diluted in the reaction buffer, and the DNA substrate was added, followed by a 15 min incubation at 37 °C. The same procedure was performed using DMSO as a control under identical experimental conditions.

### 4.4. Religation Kinetics

Analyses of religation kinetics were carried out by using DNA Oligonucleotides CL14-FITC (5′-GAAAAAA**G**ACTTAG-3′); the nucleotide labeled with FITC is guanine and is represented in bold. The complementary strand of this oligonucleotide, namely CP25 (5′-TAAAAATTTTTCTAAGTCTTTTTTC-3′), was phosphorylated with ATP at its 5ʹ- end. The CL14 strand of the oligonucleotide was annealed with a 2-fold molar excess of CP25 to obtain partially duplex CL14/CP25 suicide substrate (SS). This SS was incubated with 1.2 pmol hTOP1 (Prospec) for 60 min at 25 °C, then for an additional 30 min at 37 °C in 20 mM Tris HCl pH 7.5, 0.1 mM Na2EDTA, 10 mM MgCl_2_, 50 μg/mL acetylated BSA, and 150 mM KCl. After the formation of the cleavage complex (CL1), a 5 μL aliquot was removed, representing time 0. In the next step, 200-fold molar excess of R11 oligonucleotide (5′-AGAAAAATTTT-3′) over duplex CL14/CP25 was added to initiate religation kinetics. Aliquots containing 5 μL were removed at different time points and the reaction was stopped by addition of 0.5% SDS. The samples were then subjected to ethanol precipitation. After this the samples were digested by adding 5 μL of 1 mg/mL of trypsin and incubated at 37 °C for 60 min. Since trypsin does not digest hTOP1 completely, there is a trypsin resistant peptide that remains attached to the substrate causing the 12nt (CL1) oligo to run slower than the uncleaved band in the gel. The samples were then analyzed by using 7 M urea/20% denaturing polyacrylamide gel electrophoresis running buffer containing 48 mM Tris, 45.5 mM Boric Acid, and 1 mM EDTA. Gels were visualized using a ChemiDoc MP system (Bio-Rad Laboratories, Hercules, CA, USA) and analyzed by densitometry with Image J software 6.1.

### 4.5. HTOP1-Mediated DNA Cleavage Reactions

DNA cleavage reactions, except for the DNA substrate, were prepared as previously reported [25]. Briefly, a 48-bp DNA oligonucleotide substrate was designed by combining the previously identified TOP1 cleavage sites with modifications. The oligo had three primary resources: H-15826, previously used pBluescript SK (-) site, and D3. H-15826 was the site at human mitochondrial DNA position 15826 on the heavy strand [26]. D3 was the site from DiGate and Marians, [27]. The DNA oligo was 3′ end-labelled by 6-FAM. After 20 min of incubation at 25 °C with 1µM of CPT, 10 µM PA, or both CPT and PA, 0.35 M NaCl was added to inhibit the formation of new hTOP1 cleavage complexes to study the reversal (relegation) rate of the hTOP1 cleavage complexes. Aliquots were taken and reactions were terminated at various time points. Reactions were terminated by adding 1 volume of gel loading buffer [99.5% (*v*/*v*) formamide, 5 mM EDTA] and analyzed through a 20% denaturing PAGE. Gel images were scanned using a Typhoon FLA 9500 scanner (GE Healthcare), and densitometry analyses were performed using ImageQuant software (https://www.cytivalifesciences.com/en/us/shop/protein-analysis/molecular-imaging-for-proteins/imaging-software/imagequant-tl-analysis-software-p-28619?srsltid=AfmBOor-IoX92ySV8POawbnM5I1zf0pFQ7KVwQeyIWgeRWRBEXS1YVKn, accessed on 21 February 2025) (GE Healthcare).

### 4.6. Molecular Modeling

The crystal structure of the hTOP1 (PDB ID: 1T8I, chain D) [28], lacking the N-terminal domain (residues 1–200), has been used to derive a hTOP1 representing the transitory phospho-tyrosine link phase (named covalent hTOP1). The gap between nucleotides dT10 and dG11 was repaired. The tleap module of the AmberTools22 program [29], the parmbsc1 force field [30], and the Chimera software [31] were used to reconstruct the two nucleotides and the missing phosphodiester bond.

The structure of the CPT was retrieved from the same PDB [28]. The structure of the PA (PubChem CID: 985) was retrieved from the PubChem compound database [32].

A system composed of hTOP1-DNA, one molecule of CPT, and ten molecules of PAs was built. The DynamicBind program [33] identified three druggable sites of PAs onto the hTOP1 (Appendix A). The remaining seven PA molecules, required to mimic the experimental concentration, were added to the simulation system using the Packmol software [34]. The Chimera program [31] was finally used to assemble the three simulation systems, namely the hTOP1-DNA, the hTOP1-DNA-CPT, and the hTOP1-DNA-CPT-PAs complexes.

To computationally mimic the hTOP1 religation assay, a system composed of a linear DNA and ten PA molecules was built. The DNA sequence used corresponds to the R11 oligonucleotide (5′-AGAAAAATTTT-3′), which is the target sequence in the religation assay. The AlphaFold3 web server [35] was used to model the interaction between the DNA and PA molecules and retrieve a combined initial structural model.

### 4.7. Molecular Dynamics Simulations

Topologies of the complex structures obtained from the molecular modeling were generated using the tleap module of the AmberTools22 program [29]. AMBER ff19SB [36] and parmbsc1 [30] were used to parametrize the hTOP1 and DNA. At the same time, ligand parameters were generated using the antechamber module of the AmberTools22 program [29] and the general Amber force field [37]. Each complex was inserted in a box of TIP3P water molecules and 0.15 mol/L of NaCl [38].

Five minimization cycles were performed to remove unfavorable interactions, each gradient minimization. A starting restraint of 20 kcal·mol^−1^·Å^−2^ was imposed on the protein, DNA and ligand atoms; it was then slowly reduced and removed in the last minimization cycle. The system’s temperature was gradually increased from 0 to 300 K in an NVT ensemble using the Langevin thermostat [39] over a period of 2.0 ns. A starting restraint of 0.5 kcal·mol^−1^·Å^−2^ was imposed on the protein, DNA, and ligand atoms, then gradually decreased to slowly relax the system. Systems were then simulated in an isobaric–isothermal (NPT) ensemble for 2.0 ns using the Langevin barostat [40], imposing a pressure of 1.0 atm and maintaining the temperature to 300 K. The SHAKE algorithm [41] was used to constrain covalent bonds.

For hTOP1-DNA, hTOP1-DNA-CPT, and hTOP1-DNA-CPT-PAs systems, classical MD simulations were performed with a production run of 500 ns using the pmemd.cuda module of AMBER22 [42], with a timestep of 2.0 fs. On the contrary, three independent replicas of 250 ns each were performed using GaMD simulations for the systems composed of the linear DNA and PAs mimicking the religation assay. System coordinates were saved every 1000 steps for all simulations to capture the dynamics over time. Long-range interactions were calculated using the PME method [43], while a cut-off of 9.0 Å was imposed for short-range interactions.

### 4.8. Trajectory Analysis

To analyze the collective motions of hTOP1 in the three simulated systems, we applied principal component analysis (PCA) on the Cα atoms using GROMACS 2022.2 [44]. This involved diagonalizing the covariance matrix generated for each trajectory using the GROMACS *covar* module [44]. The covariance matrix was constructed from the atomic fluctuations of Cα atoms after removing translational and rotational movements. Additionally, we calculated the radius of gyration using the *gyrate* module and the buried surface area using the *sasa* module of GROMACS [44], supplemented by an in-house Python script. Finally, we have performed molecular mechanics/Poisson–Boltzmann–Born surface area (MM/PBSA) analyses [45] on each trajectory using the MMPBSA.py.MPI program (AmberTools22) in AMBER22 [29] with an ionic strength of 0.15 M.

## Figures and Tables

**Figure 1 ijms-26-02018-f001:**
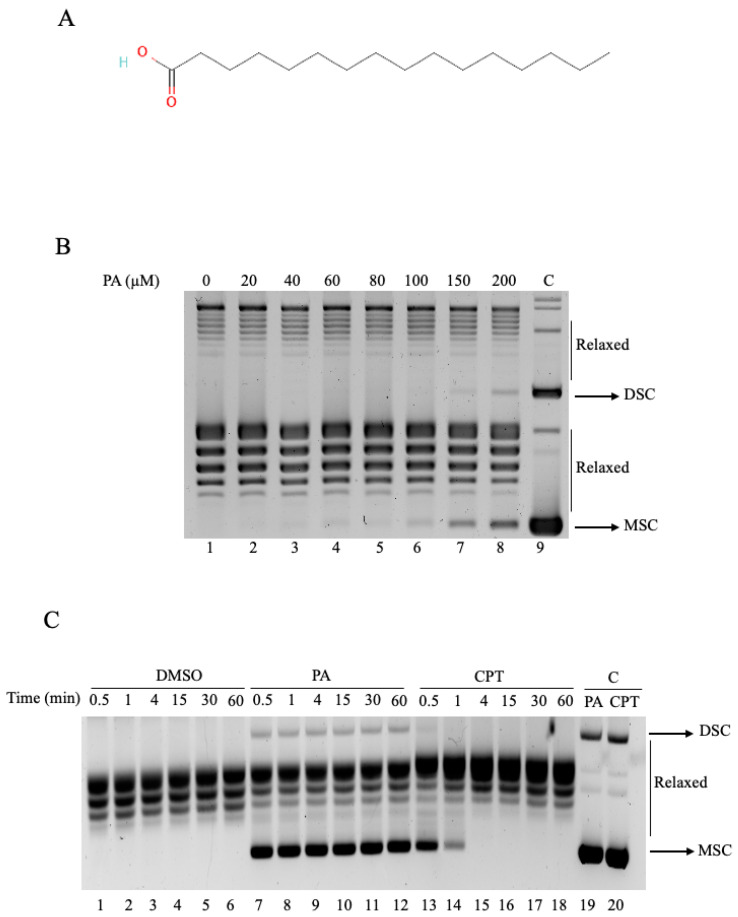
Relaxation of supercoiled DNA in presence of PA. (**A**) 2D structure representation of PA. (**B**) Relaxation of negative supercoiled DNA plasmid by hTOP1 at increasing PA concentrations (lanes 2–8), lane 1 DMSO and lane 9 with 200 μM PA and no protein added. (**C**) Relaxation of negative supercoiled DNA plasmid in a time course experiment with DMSO (lanes 1–6), with 100 µM PA (lanes 7–12), and 100 μM CPT (lanes 13–18); lanes 19 and 20 correspond to samples with 100 µM PA and 100 µM CPT, respectively, with no protein added. Reaction products were resolved on agarose gel and visualized with ethidium bromide (EtBr). DSC—dimer supercoiled DNA plasmid; MSC—monomer super-coiled DNA plasmid; C—negative control (corresponding to samples with 100 µM PA and 100 µM CPT, respectively, with no protein added).

**Figure 2 ijms-26-02018-f002:**
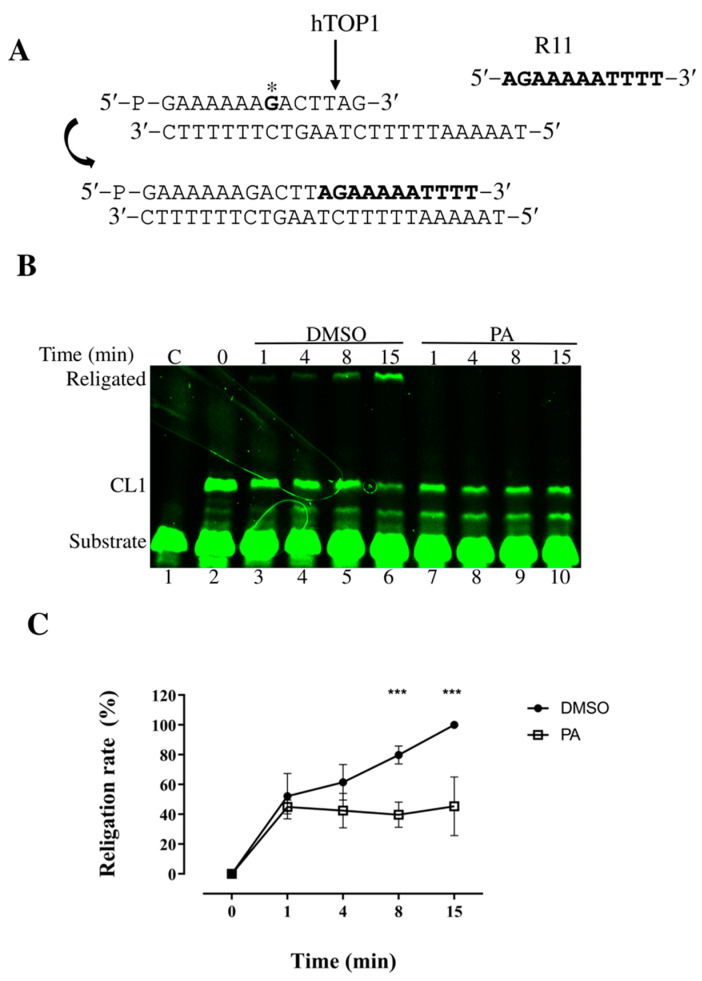
Analysis of religation of hTOP1 catalytic mechanism using FITC (fluorescein isothiocyanate) oligonucleotide labeled SS. (**A**) Top panel displays sequences of fluorescently FITC labeled SS used in religation assay, asterisk indicates that FITC was conjugated to guanine. (**B**) Representation of a denaturing polyacrylamide gel of the religation assay. Samples were incubated for 1 h at 25 °C followed by 30 min at 37 °C. Reaction was initiated by adding a 200-fold excess of R11 oligonucleotide, either with or without 100 µM PA, then stopped at various time points with 0.5% SDS. CL1 represents cleaved strand (TOP1cc), C is negative control (no protein added), and 0 denotes TOP1cc starting condition before addition of R11. (**C**) Plot illustrates percentage of religated bands over time from religation assay. Figure presents cumulative data with mean ± SD from three independent experiments. Statistical significance is indicated with asterisks: *** *p* ≤ 0.001.

**Figure 3 ijms-26-02018-f003:**
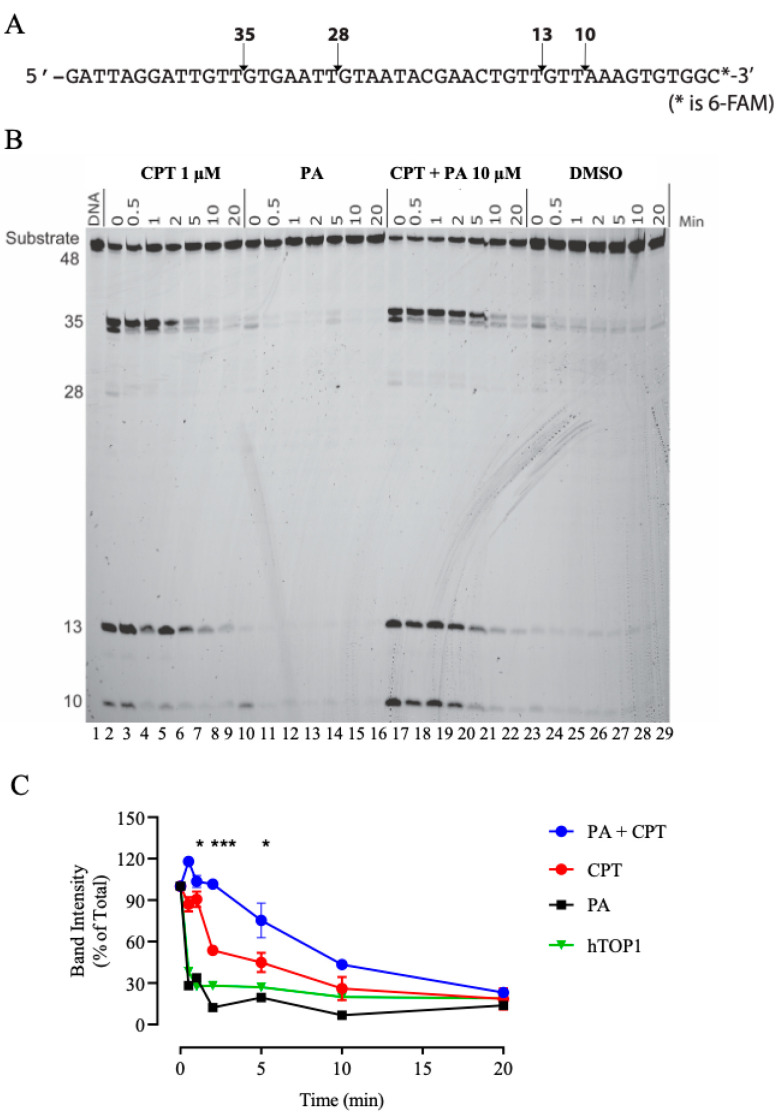
hTOP1–DNA cleavage complex reversal assay. (**A**) Top panel displays sequences of fluorescently labeled SS used in the assay. (**B**) Polyacrylamide gel reporting kinetics of formation of PA and CPT induced hTOP1-mediated DNA cleavage complexes. 3′-6-FAM end labeled 48 bp oligonucleotide was reacted with hTOP1 in presence or absence of 1 µM CPT, 10 µM PA, or both at 25 °C for 20 min. DNA cleavage was reversed by adding 0.35 M NaCl and monitored over time. (**C**) Graph reporting 35 bp band quantification as function of time for PA and CPT (blue line), CPT (red line), PA (black line), and hTOP1 (green line) as control. Samples are represented as mean value ± SD. * *p* < 0.05 *** *p* < 0.001.

**Figure 4 ijms-26-02018-f004:**
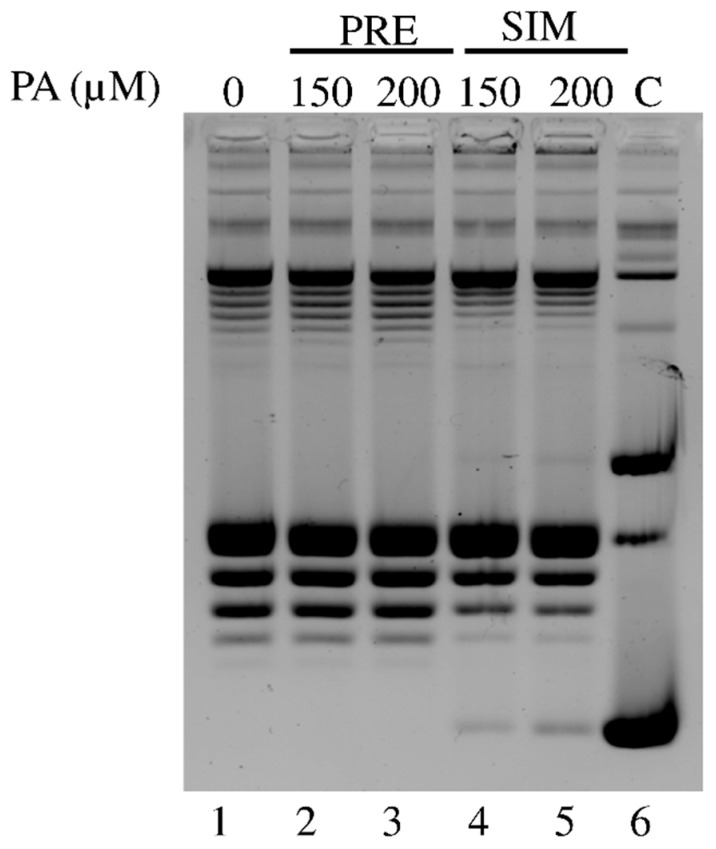
Pre-incubation dose-dependent relaxation assay. Relaxation of negative supercoiled plasmid DNA in a dose-dependent experiment with DMSO (lane 1), 150 μM and 200 μM PA in pre-incubation condition, indicated as PRE (lanes 2–3), and 150 μM and 200 μM PA in simultaneous condition, indicated as SIM (lane 4–5), with no protein added in lane 6. Reaction products are resolved on agarose gel and visualized with EtBr. C indicates negative control.

**Figure 5 ijms-26-02018-f005:**
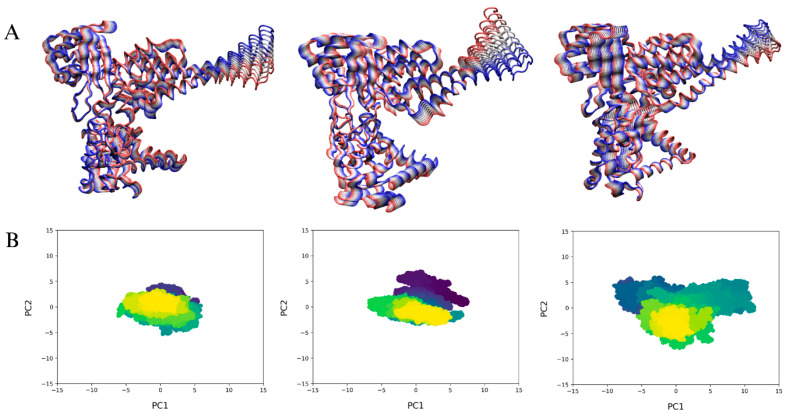
Essential motions of MD simulations. (**A**) Representation of two extreme projections of motions described by first eigenvector (PC1), interpolated onto 3D structures of hTOP1-DNA (left), hTOP1-DNA-CPT (center) and hTOP1-DNA-CPT-PAs (right) systems. Direction and amplitude of the internal motions are shown as color shift from blue to red and width of ribbons, respectively. (**B**) 2D projections of first (PC1) and second (PC2) eigenvectors of hTOP1-DNA (left), hTOP1DNA-CPT (center) and hTOP1-DNA-CPT-PAs (right) systems. Color coding shows progression from starting (violet) to final (yellow) stages of the simulations.

**Figure 6 ijms-26-02018-f006:**
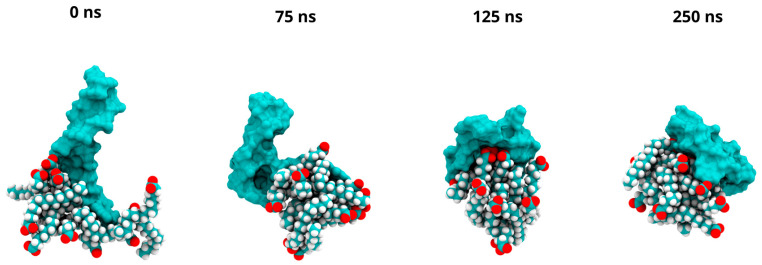
MD of PA-DNA systems. Representative snapshots for first replica are shown at 0 ns, 75 ns, 125 ns, and 250 ns. For each figure, DNA is shown using a cyan surface representation, while PA molecules are shown in Van der Waals representation.

**Table 1 ijms-26-02018-t001:** Result of MM/PBSA analyses of MD trajectories of hTOP1-DNA-CPT and hTOP1-DNA-CPT-PAs complexes.

System	Interaction Energy (kcal/mol)	Electrostatic (kcal/mol)	VdW (kcal/mol)
hTOP1-DNA-CPT	−33.08 ± 3.86	−19.39 ± 8.83	−49.02 ± 3.07
hTOP1-DNA-CPT-PAs	−43.45 ± 3.39	−29.83 ± 7.80	−53.77 ± 2.97

## Data Availability

The original contributions presented in this study are included in the article/Appendix A. Further inquiries can be directed to the corresponding author.

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
