# Peer review of "Unveiling the Mechanism of Action of Palmitic Acid, a Human Topoisomerase 1B Inhibitor from the Antarctic Sponge *Artemisina plumosa"

_ijms, 2025, doi:10.3390/ijms26052018_

Round 1

Reviewer 1 Report (Previous Reviewer 2)

Comments and Suggestions for Authors

The authors provided the proper corrections. The manuscript is almost accepptable if the authros further correct the minor points below.

1. The introduction paragraph is too long. Please divide it to the sevral paragraphs.

Author Response

  1. The introduction paragraph is too long. Please divide it to the several paragraphs.

 Thank you for your valuable feedback. We appreciate your comment regarding the introduction, and we have now divided it into multiple paragraphs to improve readability and clarity.

Reviewer 2 Report (Previous Reviewer 1)

Comments and Suggestions for Authors

The revised manuscript by Ottaviani et al titled ‘Unveiling the Mechanism of Action of Palmitic Acid, a Human Topoisomerase 1B Inhibitor from the Antarctic Sponge Artemisina plumosa’ describes the potential mechanism for the human DNA topoisomerase I (TOP1) inhibitor, palmitic acid (PA), from the Antarctic Sponge Artemisina plumosa. The authors revised the manuscript to great length and responded to the reviewer’s request and commends very well. This improved the manuscript quality and supported the suggested mechanism of action of PA as hTOP1 inhibitor.

There are some minor issues that need to be addressed before I can recommend this manuscript to be accepted for publication.

Minor comments:

Fig 1b, L120-122: Provide quantitive values of the presence of supercoiled DNA in the PA treated lanes relative to the untreated lane.

Carefully read, edit the text so the main text and the method section are describing the same conditions. E.g., L180 and 191; ‘…. reactions are stopped with 2.5% SDS.’ and ‘…. various time points with 2.5% SDS.’, while it is 0.5% in the method sections.

Although, the authors acknowledge that there is no structure of full-length hTOP1 and as such they can not model the potential interactions between PA and the N-terminal domain residues of hTOP1, they can biochemically determine if the N-terminal domain of hTOP1 is involved in the mechanism of action PA which is the focus of this manuscript. It is the latter that is missing in the manuscript described studies, yet an important contribution to the potential mechanism of how PA inhibits hTOP1. Especially as their molecular modeling/docking of the PA with the TOP70 structure of hTOP1 suggests multiple binding spaces for PA, and the shown potential that PA affects the topology of the DNA once bound by hTOP1. As such, it is essential to understand the role of the N-terminal domain in the mechanism of PA inhibition of hTOP1. The comparison of full-length hTOP1 with hTOP70 will provide a potential role for the N-terminal domain of hTOP1 in the mechanism of PA inhibition using the biochemical assay that are depicted in figure 1b/c, 2 and 3.

Did the authors consider that PA might have activity against type 1A topoisomerases, human TOP3x/b. These enzymes also act as a monomers as hTOP1 and have a more open conformation including a ‘C-type like clamp’ than the dimeric TOP2 enzymes.

L483; there is no asterisk depicted within the sequence of the oligonucleotide. Please, edit the sequence by for example bolding the nucleotide that is FITC labeled and remove ‘FITC’ from the sequence. As is it is not clear what is labeled.

Author Response

1 ) Fig 1b, L120-122: Provide quantitive values of the presence of supercoiled DNA in the PA treated lanes relative to the untreated lane.

Thank you for your comment. We have quantified and graphed the presence of supercoiled DNA in the PA-treated lanes relative to the untreated lane, as requested. We are attaching the graph with this response for your reference. If necessary, we are happy to include it in the figure section of the manuscript.

2) Carefully read, edit the text so the main text and the method section are describing the same conditions. E.g., L180 and 191; ‘…. reactions are stopped with 2.5% SDS.’ and ‘…. various time points with 2.5% SDS.’, while it is 0.5% in the method sections.

Thank you very much for bringing to our attention the inconsistency regarding the SDS concentration in the main text and the methods section. We sincerely appreciate your careful reading of the manuscript. The discrepancy, arising from the fact that in one instance it was referred to the stock solution used, while in the other the final concentration used, was corrected. Specifically, the SDS concentration is now reported as 0.5%, the final concentration, in both the main text page 5, line 180 (now line 185), page 6, line 191(now line 197) and page 7, line 234 and the methods section. We have made the changes and highlighted them in red for clarity.

3) Although, the authors acknowledge that there is no structure of full-length hTOP1 and as such they cannot model the potential interactions between PA and the N-terminal domain residues of hTOP1, they can biochemically determine if the N-terminal domain of hTOP1 is involved in the mechanism of action PA which is the focus of this manuscript. It is the latter that is missing in the manuscript described studies, yet an important contribution to the potential mechanism of how PA inhibits hTOP1. Especially as their molecular modeling/docking of the PA with the TOP70 structure of hTOP1 suggests multiple binding spaces for PA, and the shown potential that PA affects the topology of the DNA once bound by hTOP1. As such, it is essential to understand the role of the N-terminal domain in the mechanism of PA inhibition of hTOP1. The comparison of full-length hTOP1 with hTOP70 will provide a potential role for the N-terminal domain of hTOP1 in the mechanism of PA inhibition using the biochemical assay that are depicted in figure 1b/c, 2 and 3.

Thank you for your insightful comment. We fully understand the importance of comparing the full-length hTOP1 with the truncated hTOP70 in the context of the mechanism of PA inhibition. However, at present, we do not have rapid access to the hTOP70 protein. Therefore, we are unable to perform the requested experiments comparing the full-length and N-terminal truncated versions of the protein at this stage in reasonable time. However, we agree that such experiments would be valuable and could provide critical insights into the role of the N-terminal domain in PA inhibition. We plan to conduct these experiments in the future when we have access to the N-terminal truncated form. Additionally, we believe this could be an exciting direction for other research groups who are interested in studying the role of the N-terminal domain in the mechanism of PA inhibition. We have included a sentence in the manuscript to highlight this limitation and indicate that these experiments could be pursued in future studies (page 12, lines 425-428).

4) Did the authors consider that PA might have activity against type 1A topoisomerases, human TOP3x/b. These enzymes also act as a monomers as hTOP1 and have a more open conformation including a ‘C-type like clamp’ than the dimeric TOP2 enzymes.

Thank you for your insightful suggestion regarding the potential activity of PA against other kinds of topoisomerases, such as type 1A. We agree that it is an interesting idea to explore, and we appreciate your thoughtful consideration of this possibility. However, our current research does not focus on type 1A topoisomerases. We believe this paper could serve as a valuable starting point for other research groups studying these enzymes, and we hope it sparks further investigation in this area.

5) L483; there is no asterisk depicted within the sequence of the oligonucleotide. Please, edit the sequence by for example bolding the nucleotide that is FITC labeled and remove ‘FITC’ from the sequence. As is it is not clear what is labeled.

Thank you for pointing out the issue with the labeling of the oligonucleotide sequence. We agree that the current presentation is unclear, and we appreciate your suggestion. In response, we have removed the "FITC" from the sequence and instead bolded the nucleotide that is FITC-labeled (G) to make it clearer. Additionally, we realized that the figure 2A did not include a description of the asterisk. We have now added the explanation in the legend page 6, lines 193-194. We hope this revision improves the clarity of the used sequence. Line 483 (page 14) is now lines 487-489.

Round 2

Reviewer 2 Report (Previous Reviewer 1)

Comments and Suggestions for Authors

Please insert the generated graph of the quantified  values of figure 1B and describe in figure legend.

Author Response

1. Please insert the generated graph of the quantified values of figure 1B and describe in figure legend.

Thank you for your valuable suggestion. We have now included the generated graph of the quantified values from Figure 1B as Figure S2 in the supplementary figures. Consequently, all other supplementary figures have been renumbered accordingly, and the text has been updated to reflect these changes The corresponding description of Figure S2 has been added to the figure legend as suggested and is also mentioned in the Results section on page 3, lines 130-132.

This manuscript is a resubmission of an earlier submission. The following is a list of the peer review reports and author responses from that submission.

Round 1

Reviewer 1 Report

Comments and Suggestions for Authors

ijms-3340914-peer-review-v1

The manuscript by Ottaviani et al titled ‘Unveiling the Mechanism of Action of Palmitic Acid, a Human Topoisomerase 1B Inhibitor from the Antarctic Sponge Artemisina plumosa’ describes a potential novel human DNA topoisomerase I (TOP1) inhibitor from the Antarctic Sponge Artemisina plumosa. They present data that palmitic acid (PA) acts as an irreversible TOP1 inhibitor, preventing TOP1-DNA covalent complex (TOP1cc) to dissociate by impairing the DNA ligation step. This reviewer is not convinced that the authors show that PA is an irreversible TOP1 inhibitor, and they do not show any comparison for this claim with the reversible TOP1 inhibitor camptothecin or its clinically active analogs. Moreover, preincubation of TOP1 with PA does not impair TOP1 relaxation activity, and simultaneous incubation only prevent relaxation of DNA by TOP1 less then 5%, suggesting that PA does not prevent relaxation of the supercoiled plasmid. PA does affect TOP1 religation or prevents the added religation strand to enter the catalytic pocket as might be suggested from the authors dynamic modeling data. This reviewer is surprised that the authors do not provide evidence that PA trap or prevents the formation of TOP1cc in a band-shift/EMSA experiment. Although, the cleavage reversal experiment and the religation experiment, suggests that PA prevents TOP1 cleavage as no or additional (religation experiment) cleavage products are detected, which confuses this reviewer. Overall, the presented results suggest PA might be a catalytic inhibitor similar to the experimental TOP1 catalytic inhibitor CY13II, which activity is in a similar concentration range (100-150 uM) (Wu et al 2010). On the other hand, PA treatment with CPT seems to enhance CPT (called an TOP1 inhibitor but acts as a TOP1 poison) ability to trap TOP1cc, which mechanism needs to be explained by the authors.

The observation that PA inhibits TOP1 is not novel, as the authors mention in their discussion with a reference to two publications (Karna at al 2012 and Harada et al 2002), so it is unclear what the novelty factor of this manuscript is since the authors do not present a mechanism of action for PA ability to affect TOP1 activity. Addition of the mechanism would provide the critically needed novel element to this manuscript.

In addition, the authors do not discuss the PA treatment concentration in perspective of the human body concentration. For example, Fatima et al (2019) reports that PA is the most common saturated fatty acid in the human body, account for about 65% of saturated fatty acids, and ~30% of the total fatty acids in serum. This raises the question how TOP1 can function in the presents of such high concentration of PA? The PA concentration in the human body is easily in the range that, according to the presented data, affect TOP1 function. The authors use a 100uM PA for their TOP1 activity inhibition experiments, which is in the range of PA plasma levels of healthy individuals (100-1900 uM) (Fatima 2029). Moreover, high concentrations of PA are toxic for cell, PA is used as post-translational modification and as signal molecule.

Furthermore, the manuscript contains errors such as that ‘human TOP1B exhibits a broad range of functionality across different organism from prokaryotes to eukaryotes’ (L43-45), while the type 1B topoisomerases are found only in eukaryotic organism. The authors are confused with the type 1C to which topoisomerase V belong, which was originally classified in 1993 as a type 1B, but due to its lack of sequence homology put in its own class in 2006 (e.g. Forterre et al 2006 or reviewed in Osterman et al 2022). Furthermore, the method sections does not describe the methods used and often refer to other publications that do not describe the methods as they refer to other references. This is not done, do not send the reader on a treasure hunt without a result. Just describe the method used, and if too long add this to supplemental data. The reader needs to understand how the experiment was performed. Also, do not copy-paste from previous publications, e.g., L435-437; ‘The cleavage…. Previously [19]’, is literally copied from ref 19, and not corrected for the molecule used. ‘DM” or Dimethylmyricacene is not used in this manuscript but the subject of reference 19, a paper by these authors in Cells 2022.

In Sum, the authors need to provide convincing data that support the claim the PA is an irreversible TOP1 inhibitor, provide a potential mechanism of action, compared PA  effects to the characteristic of camptothecin (a reversible TOP1 inhibitor/poison, and an experimental TOP1 catalytic inhibitor e.g., CY13II). PA as a TOP1 inhibitor is not novel. As such, I recommend not to accept the manuscript in its present form, but to provide the authors time to generate experimental data that supports their claim and a potential mechanism of action by submitting this manuscript in with novel data.

Commends:

General:

This manuscript needs to be edited for spelling and grammar, and inaccurate descriptions besides the one mentioned as an example the commends above.

The method section needs to be expanded in description of the experimental procedure, including buffer compositions etc.

Edit figure legends by adding a description of the experiment and what is shown, so the reader can draw its own conclusions.

Specific:

L102-103; lane 9 in Fig. 1b does not show DNA with 200uM PA, please edit text or figure.

Figure 1C and text L106-117; The authors suggests that PA is irreversibly inhibiting TOP1 relaxation. First, they do not show result from a reversible TOP1 inhibitor (e.g., CPT) for comparison, second, the result suggests that PA is a catalytic inhibitor as it prevents TOP1 activity, within 30 seconds of the addition of TOP1 to the DNA/PA mixture. 30 Second time point shows already relaxation (TOP1 activity) of the supercoiled DNA which is not further relaxed suggesting catalytic inhibition. To show that this is irreversible the authors should for example remove PA from the reaction conditions and follow the relaxation activity of the DNA/TOP1 mixture.

Fig 1 legend, explain what the negative control (lane 9 and 15) represents.

Fig S1B, why is the DNA content of Lane 5 reduced?

Fig 2B: Missing in the original gel is the TOP1cc band. This should be visible at or near the wells. These TOP1ccs should not change in the PA treated lanes and should change in the DMSO treated lanes, do the authors see this? and have they checked similar levels of TOP1cc at the start for DMSO and PA treated samples?

Also, a comparison with CPT, is a critical control that is missing.

Fig 2 legend; please clearly label what is fluorescently labels, which strand is CL1, what is the negative control etc.

According to figure 3A the 6FAM label is located at the 5' end, is this correct? The legend describes the 6FAM at the 3’ end, as does the method section. Moreover, the labeling of the 10, 13, 28 and 35 cleavage sites is not accurate. The authors need to carefully check the legend, text and figures for errors.

The authors do not explain the use for the different reaction temperature between the different TOP1 activity assays, preventing comparison of the different observations.

The authors need to explain why pre-incubation of the TOP1 does not affect TOP1 activity. PA should still be interacting with TOP1 or does PA only act (very inefficiently) on TOP1cc as suggested by the preincubation of PA with DNA (figure 4). This result suggests that PA might interact with DNA (intercalation or by other means) to affect TOP1 activity, poorly, as more than 95% of the DNA is relaxed by TOP1.

Did the authors do TOP1 docking studies to detect potential PA binding spots or with DNA? Also, the authors do not comment on the role of the missing N-terminal domain of TOP1 as they use the only available structures of TOP1 which are N-terminal deleted proteins aka TOP70. The authors could repeat the religation and cleavage assays with a hTOP70 construct. This would potentially support the MD results and provide Fidelity to the MD results and the role of the TOP1 N-terminal domain in the effects of PA on TOP1 activity inhibition. Did the authors test the interaction between TOP1cc with and without CPT to detect PA interaction with DNA and/or protein? This could provide hints to a potential mechanism of action for PA inhibition of TOP1.

There is no reference to figure S3

This manuscript provides some mechanistic inside but lacks the cellular effects which are important to detect with e.g., RADAR or ICE to detect TOP1cc stabilization in cells by PA. In addition, the referenced manuscripts do not show that the observed effect in cells are TOP dependent (This reviewer has no access to ref 28). So, addition of a cell experiment with and without TOP1 expression (shRNA knockdown or knockout cells) would support PA as a potential clinical TOP1 inhibitor/poison.

In the discussion the author claim that PA enhances trapping of TOP1 by CPT (L318-319), data that the authors do present, or that PA is not able to trap TOP1cc by itself, for which the authors do not provide data. Please, add experimental data via EMSA, SDS-PAGE of a TOP1 activity assay with or without CPT, PA, and DMSO to support these claims.

Moreover, in the discussion the authors claim that PA needs a TOP1cc to exert its inhibitory activity. Even if this is the case, for which no supportive data is presented, PA is a poor TOP1 inhibitor as all the presented data shows efficient (>90%) TOP1 mediated relaxation of the substrates. Please, provide supportive data for this claim.

Why is an NMR experiment included in the method section as no NMR data is presented in the manuscript.

In addition, why is EDTA and 100-fold excess of MgCl2 (relative to EDTA) together in one buffer? EDTA is a scavenger of Mg2+ ions.

Reviewer 2 Report

Comments and Suggestions for Authors

Abstracting:

The authors aimed to identify anticancer agents derived from marine sources in Antarctica. Through their screening process, they isolated a sponge species, Artemisina plumosa. From its extract, the authors separated an active fraction, with palmitic acid (PA) as its main component. PA has previously been reported to exhibit inhibitory activity against TOP I. The authors further investigated this activity using both in vitro and in silico methods. While the data are presented clearly, the description in the manuscript is insufficient for a complete and accurate understanding of their findings.

1. Major Points

1.1. Correspondence Between Purpose and Conclusion

The title of the manuscript suggests that palmitic acid was extracted from Artemisina plumosa, but the Methods section states that PA was purchased from SIGMA. Additionally, while the authors describe their screening process in the Introduction and Discussion sections, it remains unclear whether the extract of A. plumosa itself exhibited antitumor and TOP I inhibitory activities, whether PA was identified as a component of the active fraction, or at what concentration. The screening process and the identification of PA should be clarified and included in the Results section.

1.2. Proper Introduction

Although the authors reference prior studies on PA in the Discussion section (Refs. [27] and [28]), it is inappropriate to imply that this manuscript is the first to identify PA as a TOP I inhibitor. These references should be introduced earlier in the manuscript, particularly in the Introduction. Proper acknowledgment of previous research will not detract from the manuscript's primary focus, which is the mechanistic investigation of PA as a TOP I inhibitor.

1.3. Figure Readability

In Figure 4, the abbreviations "PRE" and "SIM" should be clearly defined in the figure legend.

1.4. Figure Reliability

No issues were identified.

1.5. Missing or Unreferenced Figures

No issues were identified.

1.6. Sufficient Explanation of Results

No issues were identified.

1.7. Sufficient Discussion

  1. The authors state in Line 306: "the results of the time-course relaxation assays suggest that enzyme inhibition by PA is irreversible." It is unclear how the results in Figure 1C support this conclusion. Please elaborate on the reasoning. For example, can reversibility be assessed by removing PA from the reaction mixture?

  2. In Line 359, the authors mention: "Docking of PA on the binary DNA-hTOP1 complex showed multiple possible binding sites, both in the vicinity of the catalytic cleft and distal from it." Which figure corresponds to this description? Please clarify.

2. Minor Points

2.1. Typos

No issues were identified.

3. Reference Check

3.1. Percentage of Recent Works

Of the 49 references cited, only 6 (12.2%) are from publications after 2020. To strengthen the manuscript, it is recommended to include more recent studies (≥30% of the references).

3.2. Proper Referencing

Some references appear to be missing, such as Refs. 3, 9, 12, and 20. Please verify and ensure all references are correctly included.